# Alleviation of Alcoholic Fatty Liver by *Dendrobium officinale* Flower Extracts due to Regulation of Gut Microbiota and Short-Chain Fatty Acids in Mice Exposed to Chronic Alcohol

**DOI:** 10.3390/foods12071428

**Published:** 2023-03-28

**Authors:** Jingchi Zhang, Jiakun Fan, Hui Luo, Zhengwei Liang, Yanhui Guan, Xin Lei, Nianguo Bo, Ming Zhao

**Affiliations:** 1National-Local Joint Engineering Research Center on Gemplasm lnnovation & Uilization of Chinese Medicinal Materials in Southwest China, The Key Laboratory of Medicinal Plant Biology of Yunnan Province, Yunnan Agricultural University, Kunming 650201, China; 2Yunnan Characteristic Plant Extraction Laboratory, Kunming 650106, China; 3College of Tea Science, Yunnan Agricultural University, Kunming 650201, China

**Keywords:** *Dendrobium officinale* flower extract, alcoholic fatty liver disease, gut microbiota, short-chain fatty acids

## Abstract

Alcoholic fatty liver disease (AFLD) is caused by long-term heavy alcohol consumption; therefore, useful and practical methods for the prevention of AFLD are urgently needed. The edible flower of *Dendrobium officinale* contains diverse flavonoids, and has shown antioxidant activity as well as antihypertensive and anti-inflammatory effects. In this study, an AFLD model was established, the protective effect of *D. officinale* flower (DOF) ethanol extract on AFLD was evaluated, and its mechanisms were investigated by analyzing gut microbiota and short-chain fatty acids (SCFAs). DOF extract (DOFE) supplementation promoted alcohol metabolism, restored hepatic antioxidant capacity, alleviated oxidative stress, reduced inflammatory factor levels, and inhibited dyslipidemia induced by alcohol intake in chronic alcohol-exposed mice, especially in the high DOFE group. Moreover, DOFE supplementation increased the diversity, structure, and composition of the gut microbiota in mice, restored some of the abnormal SCFA levels caused by AFLD, and helped restore intestinal function. DOFE supplementation significantly increased the relative abundance of *Akkermansia*, suggesting that *Akkermansia* may be a potential target of the protective effect of DOFE. Therefore, DOFE supplementation to improve the composition of the gut microbiota may be an effective therapeutic strategy for the prevention of AFLD.

## 1. Introduction

Ethanol is transformed into acetaldehyde in the liver, and the highly toxic acetaldehyde is further metabolized into acetate, which is rapidly released into the blood thereafter. In addition, ethanol metabolism in the liver also produces highly active molecular fragments, resulting in oxidative stress and liver damage [1]. Worldwide, alcohol consumption is a major factor in preventing liver disease [2]. Types of alcohol-induced liver injury include fatty liver [3], fibrosis, and alcoholic hepatitis, and these lesions can occur individually, concurrently, or consecutively in the same patient [4]. Alcoholic fatty liver disease (AFLD) is caused by long-term heavy alcohol consumption, and is the most prevalent and curable stage of alcoholic liver disease [5]. At present, alcoholic fatty liver still has no effective treatment available in the clinic [6]. To date, the most effective treatment for AFLD is long-term abstinence from alcohol [7,8], and there are no significant advancements in alternative therapies [9]. Therefore, useful and effective methods for the prevention of AFLD have attracted considerable attention.

Around 100 trillion microorganisms make up the complex community of the human gut microbiota, which provides important physiological functions and is an essential part of the activities of human life [10,11,12]. Long-term alcohol consumption can not only lead to liver damage through alcohol-mediated liver cell damage and lipid metabolism changes, but it can also increase intestinal permeability and change the composition of the gut microbiota [13]. Both bacterial products and bacteria can then move from the intestine to the liver due to by intestinal diseases, leading to alcohol-related liver disease [14,15,16]. On the other hand, it has also been found that AFLD can be alleviated by regulating the gut microbiome through fecal microbiome transplantation [17].

Neurotransmitters are produced by microorganisms, and dietary fiber fermentation by bacteria results in the release of short-chain fatty acids (SCFAs), which are metabolites with potential neural activity [18]. SCFAs have a variety of biological functions, affecting intestinal movement, enhancing intestinal barrier function and host metabolism, and regulating the immune system [19]. In previous studies, changes in the structure of the gut microbiota can lead to changes in the content of SCFAs, thus altering intestinal barrier function [20]. Therefore, we propose that the content of SCFAs could be altered by affecting the gut microbiota to alleviate AFLD.

In recent years, early intervention and natural dietary products with strong antioxidant and anti-inflammatory properties have been paid more and more attention for the prevention and management of AFLD [21], such as dihydroartemisinin [22], *Morinda citrifolia* [23], and Dendrobium, which has been confirmed to protect liver activity [24]. *Dendrobium officinale* Kimura et Migo (*D. officinale*), a perennial herb of the orchid family, is a traditional Chinese medicine and functional food [25], widely distributed in Australia, China, the United States, and Southeast Asia [26]. The flowers of *D. officinale* contain amino acids and flavonoids, especially quercetin, kaempferol, and their derivatives, as well as phenylpropanoids, C-glycosylflavones, and O-glycosylflavones [27,28]. Recently, local food safety standards for the *D. officinale* flower (DOF) were implemented in Fujian (DBS35/001-2020), Guangxi (DBS45/062-2019), Zhejiang (DB33/3011-2020), Guizhou (DBS52/045-2020), and Yunnan provinces (DBS53/030-2021), China. Previous studies have shown that *D. officinale* flower extract (DOFE) can relieve depression in mice; it has high antioxidant activity [29], an obvious antihypertensive effect [30], and good cardiovascular-protective, anti-cancer, neuroprotective, anti-osteoporosis, and anti-inflammatory effects [31]. As the edible flowers contain amino acids, flavonoids, and fatty acids, and show antioxidative and anti-inflammatory activity, we suggest that DOFE may alleviate liver injury. In this study, an AFLD model was established, the protective effect of DOF ethanol extract on AFLD was evaluated, and its mechanisms were investigated by analyzing gut microbiota and SCFAs.

## 2. Materials and Methods

### 2.1. Preparation of Dendrobium officinale Flower Extract

*Dendrobium officinale* flower (DOF) was collected at the Longling Institute of Dendrobium (98.696° E, 24.593° N) in May 2019. One kilogram of *D. officinale* flower was extracted with 15 L 50% ethanol for 48 h. Using a rotary evaporator and vacuum, the extraction solution was filtered and concentrated to 1/5 volume at 45 °C. For subsequent usage, the concentrates were powdered after being freeze-dried, and they were kept at −80 °C.

### 2.2. Animals and Experimental Design

Eight-week-old male C57BL/6 J mice were purchased from Saiye (Gu’an) Biotechnology Co., Ltd. (Langfang, China). All mice were housed in a specific-pathogen-free (SPF) environment with a 12-h light/dark cycle, kept at a temperature of 22 ± 0.5 °C degrees. They were given free access to a chow diet and water. Moreover, all animal experimental protocols were performed with approval from the Department of Laboratory Animal Science at Kunming Medical University. According to a previous report, the AFLD mouse model was developed using the Lieber–DeCarli liquid diet, which was purchased from TROPHIC Animal Feed High-tech Co., Ltd. (Nantong, China) [32]. This diet contains 4% (*w*/*v*) ethanol, which accounts for 28% of the total calories. All mice were fed the Lieber–DeCarli control liquid diet for 5 days to adapt to the liquid diet after 1 week of acclimatization. The mice were then randomly divided into a control group (five mice) and the ethanol-fed groups (fifteen mice), and were exposed to alcohol for 6 days according to body weight. Thereafter, mice in the ethanol-fed groups were further divided into four groups (five mice in each group), three of which received DOFE supplements and one of which received the liquid Lieber–DeCarli ethanol diet. Throughout the entire experiment, the control mice were fed a Lieber–DeCarli control liquid diet. On the 11th day, mice in all DOFE supplementary groups were administered 15 mg/mL DOFE in group L (low dose), 30 mg/mL DOFE in group M (median dose), and 60 mg/mL DOFE in group H (high dose) for 4 weeks. Sterile distilled water was given to the control and model groups for 4 weeks. All of the mice underwent anesthesia after a 24-h fast at the end of the experiment. To acquire serum for biochemical analysis, samples of blood were collected from the ophthalmic venous plexus and centrifuged at 4000× *g* for 10 min. In addition, the mice were sacrificed, and their liver tissues were collected for further research.

### 2.3. Determination of Serum Biochemical Indicators

Using the corresponding kits and an automated biochemistry analyzer (Shandong Biobase Biodustry, Jinan, China), the serum lipid profile of triglyceride (TG) and total cholesterol (TC), as well as the liver function biomarkers alanine transaminase (ALT) and aspartate transaminase (AST), were determined.

### 2.4. Hepatic Histopathological Evaluation by Hematoxylin–Eosin (H&E) Staining

Liver samples were promptly fixed in 4% formalin after the mice were sacrificed, embedded in paraffin, and sectioned. After processing, the liver tissues were then stained with H&E to evaluate liver injury brought on by chronic alcohol exposure, including infiltration of hepatocyte rearrangement, inflammatory cells, and lipid accumulation. Finally, microscopy (NIKON ECLIPSE CI, Nikon, Japan) was used to visualize the histological images.

### 2.5. Analysis of Hepatic Biochemical Indicators

All experimental groups’ liver samples were homogenized in 9 mL of pH 7.4 PBS buffer after being weighed to 1 g. To obtain the supernatant, the liver homogenate was centrifuged (2500× *g*, 4 °C, 10 min). The activities of superoxide dismutase (SOD), malondialdehyde (MDA), glutathione (GSH), glutathione peroxidase (GSH-Px), ethanol dehydrogenase (ADH), acetaldehyde dehydrogenase (ALDH), catalase (CAT), total protein (TP), and triglyceride (TG) in liver tissue were determined according to the manufacturer’s protocol (Shandong Biobase Biodustry, Jinan, China). The levels of hepatic CYP2E1; inflammatory cytokines, including interleukin-6 (IL-6); and tumor necrosis factor-α (TNF-α) were measured by enzyme-linked immunosorbent assay (ELISA) based on the manufacturer’s instructions (Jiangsu Meimian Industrial Co., Ltd., Jiangsu, China) [33,34].

### 2.6. Analysis of Gut Microbiota

According to the manufacturer’s instructions, the HiPure Stool DNA Extraction Kit (Magen, Guangzhou, China) was used to extract the total DNA of fecal microorganisms, and the V3−V4 region of bacterial 16S rDNA was amplified using the barcode 341F (CCTACGGGNGGCWGCAG) and 806R (GGACTACHVGGGTATCTAAT) primers. The amplified product quality was assessed using 2% agarose gels, purified using the AxyPrep DNA Gel Extraction Kit (Axygen Biosciences, Union City, CA, USA), and quantified using the ABI StepOnePlus Real-Time PCR System (Life Technologies, Foster City, CA, USA). Sequencing libraries were constructed using the SMRTbell TM Template Prep Kit (PacBio, Menlo Park, CA, USA). Library quality assays were performed using a Qubit 3.0 Fluorometer and FEMTO Pulse system (Agilent Technologies, Santa Clara, CA, USA). Qualified libraries were sequenced using the PacBio Sequel platform. The clean tags were clustered into OTUs (operational taxonomic units) based on ≥97% similarity, and the low-quality tags were filtered according to the filtering criteria described in the literature [35]. BLAST software was used to select the NCBI 16S rDNA database for species annotation of the results. QIIME and MUSCLE were used to analyze the microbial composition of the samples for diversity. The LEfSe method was used to identify the differentially abundant taxa. A difference was regarded as statistically significant if *p* < 0.05 and LDA score ≥3.

### 2.7. Measurement of Short-Chain Fatty Acids in Feces

SCFAs in feces were measured using the Waters Acquity UPLC system (UPLC, SHIMADZU, AB SCIEX 5500 Qtrap-MS; Shimadzu Kyoto, Japan), commissioned by Genedenovo (Guangzhou, China), as follows: 100 mg of feces samples were extracted using 800 μL of 50% acetonitrile–water, vortexed for 1 min, sonicated for 30 min at 4 °C, and then centrifuged at 12,000× *g* for 15 min at 4 °C. Then, 200 μL of supernatant was added to 100 μL of 3-NPH (200 mM) and 100 μL of EDC (120 mM; containing 6% pyridine) solution (2:1:1 *v*/*v*/*v*), vortexed for 1 min, and mixed. The reaction was then carried out at 40 °C for 1 h, during which time the reaction was shaken once every 5 min; when the reaction was complete, the supernatant was centrifuged at 12,000× *g* for 15 min at 4 °C, and the supernatant was passed through a 0.22 mM filter membrane and diluted 100 times with 50% acetonitrile–water for detection. The diluents were separated by an Acquity (location) UPLC HSS T3 (1.8 µm, 2.1 mm × 100 mm) column at 40 °C and a flow rate of 0.30 mL/min. The ESI-MSn experiments were executed on the AB SCIEX 5500 Qtrap mass spectrometer with a spray voltage of 4.2 kV in positive modes; the results of negative modes were not adopted. Curtain gas and collision gas were set at 35 and 9 arbitrary units, respectively. The temperature of the ion source was 450 °C. The final integration was performed using MultiQuant software (Version 3.0), and the content was calculated using the standard curve.

### 2.8. Statistical Analysis

Results from the experiments are shown as mean ± standard deviation (SD). All statistical analyses were conducted using one-way analysis of variance (ANOVA). Data were processed using GraphPad Prism software version 9 (San Diego, CA, USA), and *p* < 0.05 was considered as the threshold for statistical significance.

## 3. Results and Discussion

### 3.1. Alleviation of Liver Injury and Lipid Disorder by DOFE

Compared to the control (CTRL) group, the body weights of alcohol-exposed mice were reduced with or without DOFE supplementation (Figure 1A). No obvious pathological abnormalities were observed in the CTRL group following hematoxylin and eosin (H&E) staining, as shown in Figure 1B. However, mouse hepatocytes in the AFLD group showed obvious pathological changes, such as a large amount of medium and tiny lipid droplets, infiltration of inflammatory cells, and disordered cell arrangement. The morphology of hepatocytes tended to normalize after supplementation with DOFE, and was quite similar to that of the CTRL group as the concentration of DOFE increased.

The activities of ALT and AST were elevated in the AFLD group compared to the CTRL group (*p* < 0.05), and compared with the AFLD group, these enzyme activities decreased after DOFE supplementation in a dose-dependent manner (*p* ˂ 0.05) (Figure 2A,B). Serum levels of ALT and AST are sensitive markers of liver injury [36], and these enzymes demonstrated that DOFE was able to alleviate liver injury induced by alcohol intake. These findings are consistent with those of a previous study, where *Dendrobium huoshanense* polysaccharide ameliorated acute liver injury by reducing AST activity [24].

The liver is an important site for the regulation of lipid metabolism, and for these metabolites, the blood is a crucial transport medium [37]. Evidence shows that alcohol exposure leads to disturbances in hepatic lipid metabolism. Therefore, serum levels of these biochemical indicators in mice were investigated (Figure 2C–F). Compared with the CTRL group, serum levels of TC, HDL, and LDL were reduced in the AFLD group, and supplementation with DOFE reversed this trend (*p* < 0.05). Therefore, DOFE treatment was able to alleviate lipid disorders caused by alcohol intake.

### 3.2. Effects of DOFE on Alcohol Metabolism, Hepatic Antioxidant Capacity, and Inflammatory Cytokine Levels

In alcohol detoxification, the liver is the primary organ [38], and most of the ethanol is oxidized by ADH to acetaldehyde, which is subsequently oxidized by ALDH to acetic acid [39,40]. In a few cases, CYP2E1 can oxidize small amounts of ethanol to acetaldehyde [41]; however, CYP2E1 produces reactive oxygen species when oxygen is used for alcohol metabolism, which leads to oxidative stress [34,42]. As shown in Figure 3A, the expression of CYP2E1 and ALDH activity in the AFLD group was significantly increased, while ADH activity was decreased compared with the CTRL group. Supplementation with DOFE restored ADH activity and decreased the level of CYP2E1. These results suggest that DOFE enhanced alcohol metabolism by suppressing CYP2E1 expression and increasing ADH activity.

The indicators MDA, SOD, CAT, GSH, and GSH-Px respond to the antioxidant capacity of an organism and can indirectly reflect liver damage [43]. It can be seen from Figure 3B–H that, compared with the CTRL group, the MDA content was significantly increased (*p* < 0.05) and GSH activity was significantly decreased in the AFLD group, indicating that the antioxidant capacity of the organism was inhibited, and this trend was reversed following supplementation with DOFE. In addition, the contents of SOD, CAT, and GSH-Px did not show significant changes. It is hypothesized that DOFE alleviated the oxidative stress induced by alcohol exposure in the livers of mice by restoring the antioxidant capacity of the liver.

The levels of TNF-α and IL-6 in the liver increased in alcohol-exposed mice compared to control mice (*p* ˂ 0.05). DOFE effectively decreased the TNF-α and IL-6 levels which had been heightened by alcohol (Figure 3I,J). Thus, DOFE treatment improved alcohol metabolism, restored antioxidant capacity, and alleviated liver inflammation in alcohol-exposed mice.

### 3.3. Effects of DOFE on the Diversity and Structure of the Gut Microbiota in Alcohol-Exposed Mice

Multiple studies have shown that dysregulation of the gut microbiota is a major factor contributing to ALD [13]. To explore how AFLD affects the gut microbiota and the regulatory effect of DOFE on the gut microbiota of AFLD mice, 16S rRNA gene sequencing was used to analyze the composition of the fecal gut microbiota in mice. As shown in Figure 4A,B, there was no significant difference between the CTRL and AFLD groups, nor in group L after DOFE supplementation, but the Chao index of the gut microbiota was significantly increased in groups M and H (*p* < 0.05). These data suggest that supplementation with 30 mg/mL to 60 mg/mL of DOFE increased the species richness of the gut microbiota in mice.

According to the Bray–Curtis principal coordinates analysis (PCoA) plot, supplementation with DOFE significantly changed the composition of the microbial community (Figure 4C). The CTRL group was different from the DOFE-supplemented group, indicating that DOFE supplementation reshaped the gut’s microbial community. Species abundance clustering analysis, based on the phylum level and the genus level, showed that the CTRL and AFLD groups were similar to group L in terms of microbial composition, and group M was in the same cluster as group H, which was consistent with the PCoA results.

At the phylum level, Firmicutes, Bacteroidetes, Verrucomicrobia, Actinobacteria, Proteobacteria, Deferribacteres, Cyanobacteria, and Tenericutes were the most abundant bacteria in all groups of stool samples (relative abundance > 0.01%) (Figure 4D,E). Supplementation with DOFE significantly decreased the relative abundance of Actinobacteria compared to the CTRL group, and the relative abundance of Verrucomicrobia significantly increased in group H compared to the other groups.

According to the corresponding LDA score, linear discriminant analysis effect size (LEfSe) and the generated classification map were used to analyze the groups with the largest differences from the phylum to the genus in the gut microbiota, as shown in Figure 5A. Compared with the CTRL group, the *Enterococcaceae* family and *Enterococcus* genus, the *Christensenellaceae* family and *Christensenellaceae*_R_7_group genus, the *Peptostreptococcaceae* family and *Romboutsia* genus, *Anaerovorax* of the *Clostridiales* family, *Lachnospiraceae*_NK4B4_group of the *Clostridiales* family, and *Dubosiella* of the *Erysipelotrichales* family were enriched in the AFLD group. Long-term exposure to alcohol and DOFE supplementation significantly increased the relative abundance of *Akkermansia*, and significantly reduced the relative abundance of *Blautia* in group H compared to the AFLD group. In addition, compared to the CTRL group, the relative abundance of the *Lachnospiraceae*_NK4A136_group was significantly increased in group M. By degrading intestinal mucin proteins and improving intestinal barrier function, *Akkermansia* was shown to play an important role in preventing alcohol-induced liver injury and improving intestinal barrier function [44]. These results suggest that *Akkermansia* may be closely related to the therapeutic effect of DOFE on AFLD.

### 3.4. Effect of DOFE on SCFA

The gut microbiota can metabolize carbohydrates and produce SCFAs, which mainly include acetic acid, butyric acid, caproic acid, propionic acid, isobutyric acid, and isovaleric acid. These are important signals between the microbiota, the host, and metabolic substrates, and they regulate the metabolism of host cells [45,46]. Additionally, they serve as vital sources of energy for intestinal epithelial cells and regulate many of their functions, including improving the function of the intestinal barrier [19]. Therefore, SCFAs produced by the gut microbiota following alcohol exposure, as well as DOFE supplementation, were analyzed, including acetic acid, hexanoic acid, valeric acid, butanoic acid, isobutyric acid, propionic acid, and isovaleric acid. Compared with the CTRL group, hexanoic acid, valeric acid, butanoic acid, isovaleric acid, and propionic acid were significantly increased in the AFLD group, and the levels of hexanoic acid, valeric acid, and isovaleric acid were significantly decreased after DOFE supplementation, and returned to the approximate levels in the CTRL group. There were no significant differences in isobutyric acid and acetic acid between the groups (Figure 6A).

At the genus level, a heat map of the correlation analysis between the gut microbiota and SCFAs at Pearson’s level was generated (Figure 6B), and the analysis revealed that *Alloprevotella* was negatively correlated with the production of acetic acid and isobutyric acid. *Blautia* was positively correlated with the production of caproic acid. *Prevotella* was one of the dominant genera of Bacteroidetes. *Blautia* is widely distributed in mammalian feces and intestines, and it is one of the dominant genera of Firmicutes. It also has various important roles in the survival and evolution of the gut and other microenvironments [47,48]. It is hypothesized that *Alloprevotella* and acetic acid inhibit the production of isobutyric acid, and that *Blautia* promotes the production of caproic acid, which, in turn, affects the metabolic phenotype. Previous studies have found that oral administration of *Blautia wexlerae* in mice induced metabolic changes, produced anti-inflammatory effects, and was inversely associated with obesity and type 2 diabetes [49]. In this study, caproic acid was elevated in the AFLD group and decreased after DOFE supplementation, while *Blautia* was proportional to caproic acid. It was deduced that caproic acid was negatively correlated with mouse health, which is consistent with previous studies [49].

## 4. Conclusions

In summary, DOFE supplementation promoted alcohol metabolism, restored hepatic antioxidant capacity, alleviated oxidative stress, reduced inflammatory factor levels, and inhibited dyslipidemia induced by alcohol intake in chronic alcohol-exposed mice, especially in group H. Moreover, DOFE supplementation increased the diversity, structure, and composition of the gut microbiota in mice, restored some of the abnormal SCFA levels caused by AFLD, and helped to restore intestinal function. This analysis revealed that DOFE supplementation significantly increased the relative abundance of *Akkermansia*, suggesting that *Akkermansia* may be a potential target for the protective effect of DOFE, which is similar to previous research results [50]. Therefore, improving the composition of the gut microbiota through DOFE supplementation may be an effective therapeutic approach for the management and even prevention of AFLD.

## Figures and Tables

**Figure 1 foods-12-01428-f001:**
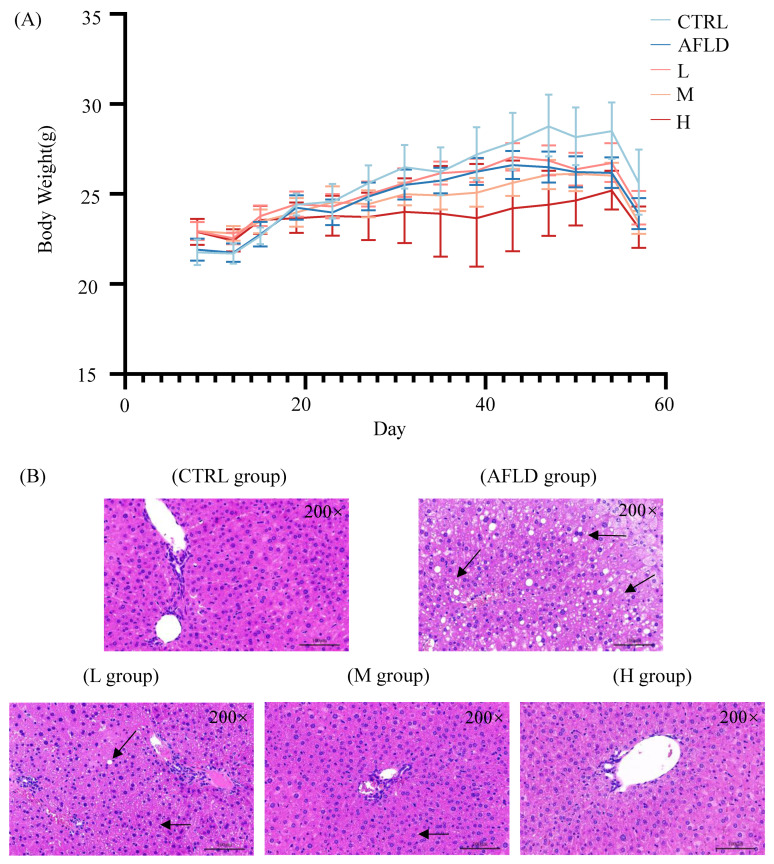
The effects of DOFE on (**A**) body weight and (**B**) representative H&E-stained liver sections from mice exposed to chronic alcohol consumption (magnification: 200, scale bar: 100 µm). CTRL, the control group; AFLD, the model group; DOFE, *Dendrobium officinale* flower extract; L, the low concentration of DOFE; M, the medium concentration of DOFE; H, the high concentration of DOFE. Black arrows point to inflammatory infiltration.

**Figure 2 foods-12-01428-f002:**
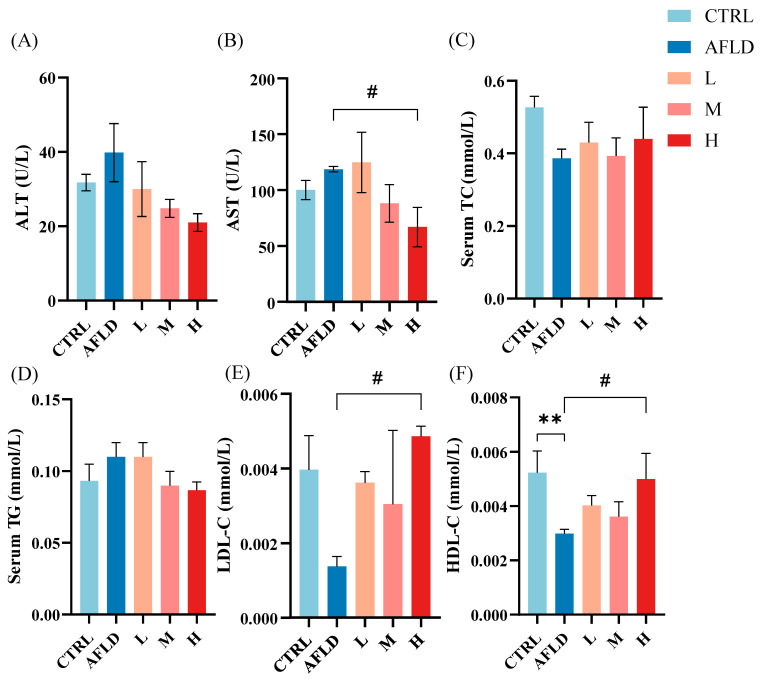
The effects of DOFE on serum biochemical markers in mice exposed to chronic alcohol consumption. (**A**) AST, aspartate transaminase; (**B**) ALT, alanine aminotransferase; (**C**) TC, total cholesterol; (**D**) TG, triacylglycerol; (**E**) HDL-C, high-density lipoprotein; (**F**) LDL-C, low-density lipoprotein. CTRL, the control group; AFLD, the model group; DOFE, *Dendrobium officinale* flower extract; L, the low concentration of DOFE; M, the medium concentration of DOFE ; H, the high concentration of DOFE. ** *p* < 0.01 for the CTRL groups compared with the AFLD groups; # *p* < 0.05 for the AFLD group compared with the DOFE high supplementary groups.

**Figure 3 foods-12-01428-f003:**
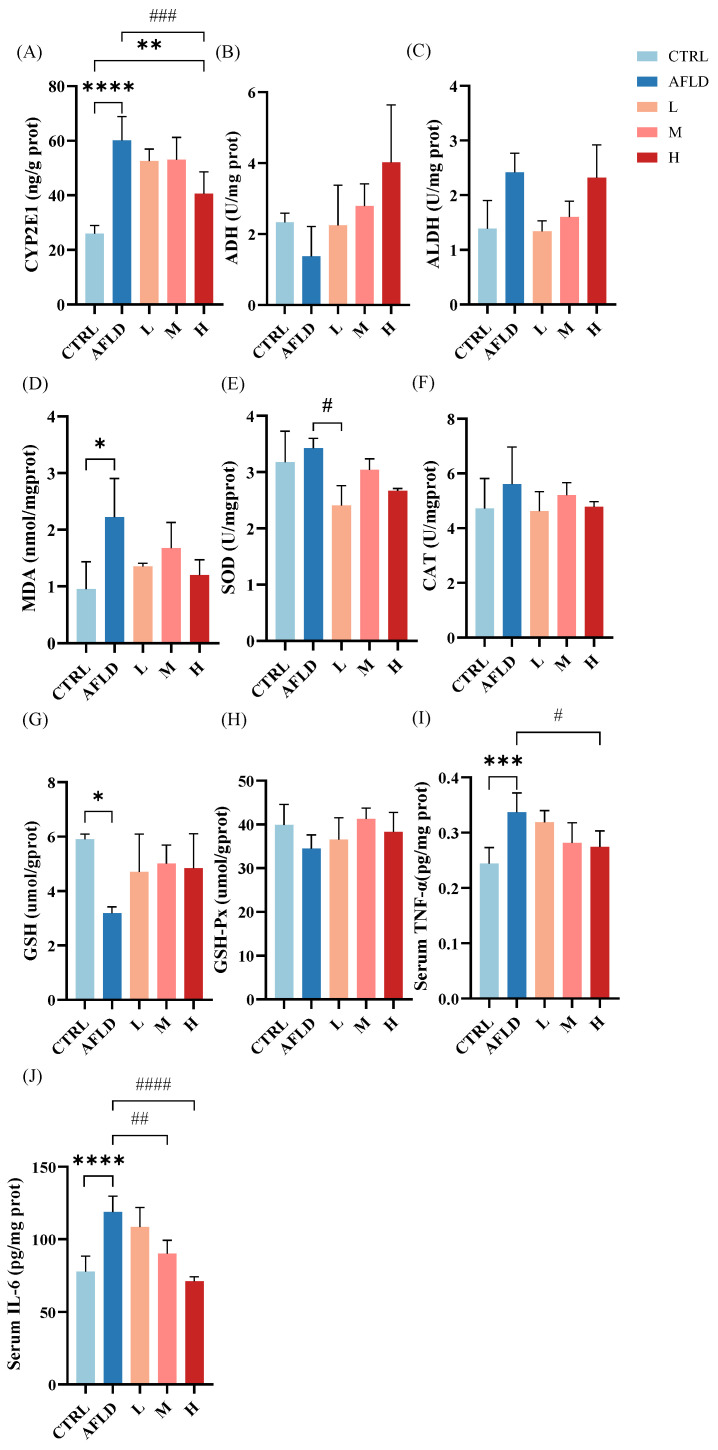
The effects of DOFE on alcohol metabolism in mice exposed to chronic alcohol consumption. (**A**) CYP2E1: cytochrome P450 2E1; (**B**) ADH: alcohol dehydrogenase; (**C**) ALDH: aldehyde dehydrogenase; (**D**) MDA: malondialdehyde; (**E**) SOD: superoxide dismutase; (**F**) CAT: catalase; (**G**) GSH: glutathione; (**H**) GSH-Px: glutathione peroxidase; (**I**) TNF-α: tumor necrosis factor-α; (**J**) IL-6: interleukin-6. CTRL, the control group; AFLD, the model group; DOFE, *Dendrobium officinale* flower extract; L, the low concentration of DOFE; M, the medium concentration of DOFE; H, the high concentration of DOFE. * *p* < 0.05, ** *p* < 0.01, *** *p* < 0.001, **** *p* < 0.0001 for the CTRL groups compared with the DOFE median supplementary groups and the DOFE high supplementary groups; # *p* < 0.05, ## *p* < 0.01, ### *p* < 0.001, #### *p* < 0.0001 for the AFLD group compared with the DOFE low supplementary groups, the DOFE median supplementary groups, and the DOFE high supplementary groups.

**Figure 4 foods-12-01428-f004:**
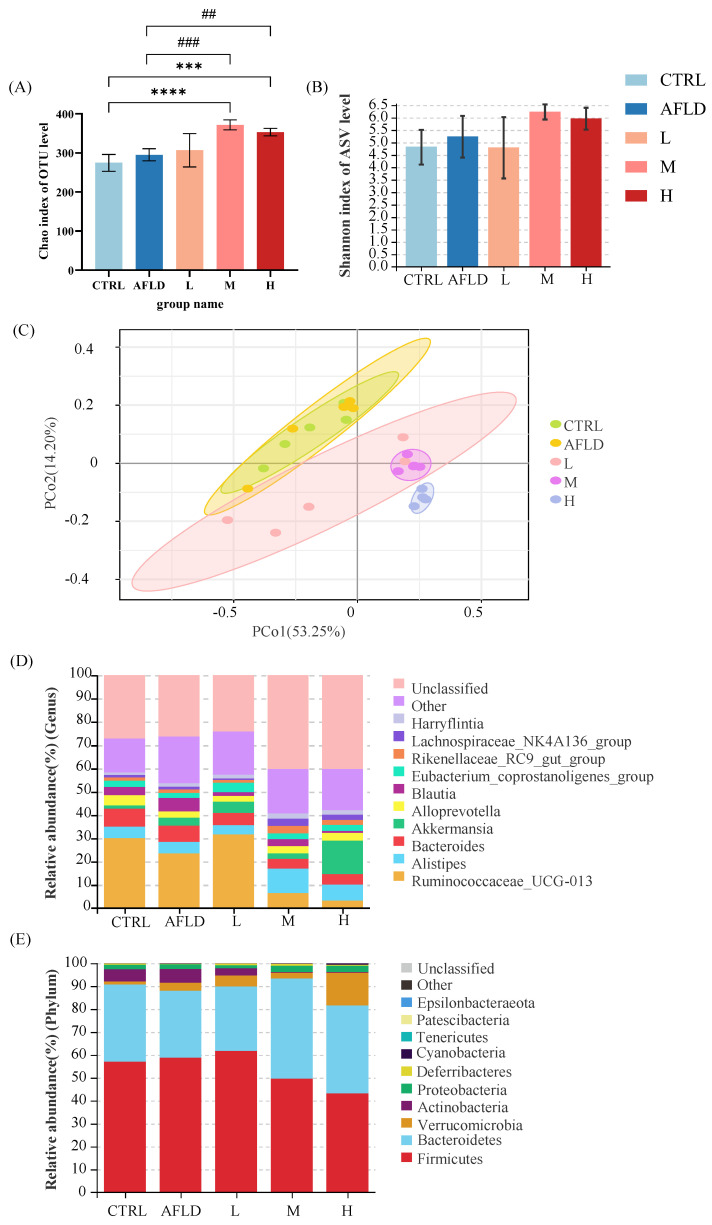
(**A**) The α-diversity representing the Chao index at the OTU level. *** *p* < 0.001, **** *p* < 0.001 for the CTRL group compared with the DOFE median supplementary group and the DOFE high supplementary group; ## *p* < 0.01, ### *p* < 0.001 for the AFLD group compared with the DOFE median supplementary group and the DOFE high supplementary group. (**B**) Shannon index to evaluate the diversity of microorganisms; (**C**) PCoA plot of the gut microbiota based on Bray–Curtis matrices. Analysis of gut microbiota composition at the phylum level (**D**) and the genus level (**E**). CTRL, the control group; AFLD, the model group; DOFE, *Dendrobium officinale* flower extract; L, the low concentration of DOFE; M, the medium concentration of DOFE; H, the high concentration of DOFE.

**Figure 5 foods-12-01428-f005:**
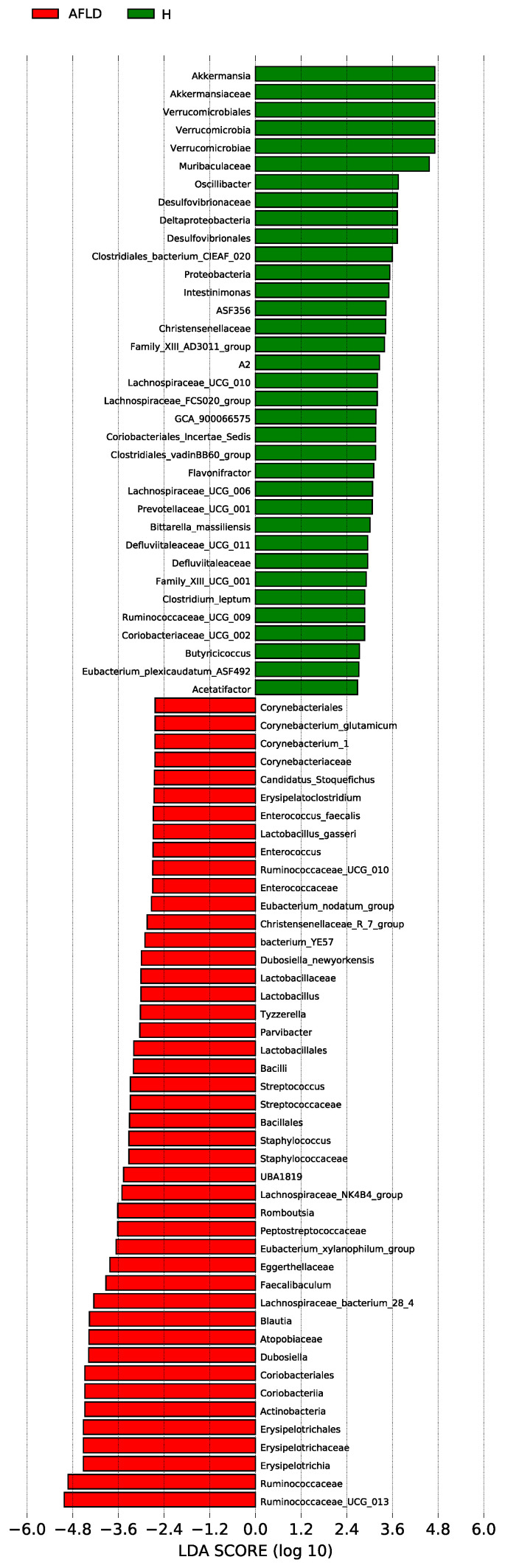
The LDA distribution chart, generated from LEfSe, shows the most differentially abundant taxa in the intestinal microbiota, ranging from phylum to genus (LDA score > 4). AFLD (red): the model group; H (green): high concentration of DOFE.

**Figure 6 foods-12-01428-f006:**
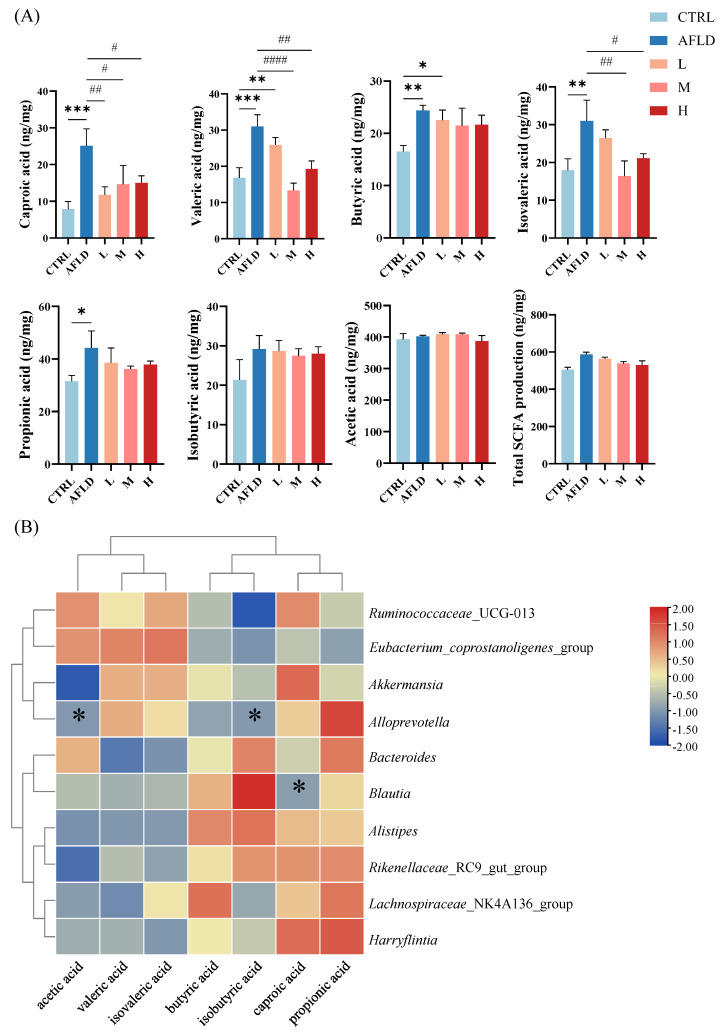
The effects of DOFE on short-chain fatty acid (SCFA) production. (**A**) The contents of SCFAs. CTRL, the control group; AFLD, the model group; L, low concentration; DOFE, Dendrobium officinale flowers extracts; M, medium concentration of DOFE; H, high concentration of DOFE. * *p* < 0.05, ** *p* < 0.01, *** *p* < 0.001 for the CTRL group compared with the AFLD group; # *p* < 0.05, ## *p* < 0.01, #### *p* < 0.0001 for the DOFE high-supplementary group compared with the AFLD group. (**B**) Heatmap of correlation analysis between gut microbiota and SCFAs (* *p* < 0.05).

## Data Availability

The data presented in this study are available on request from the corresponding author.

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
