# Peer review of "Alleviation of Alcoholic Fatty Liver by Dendrobium officinale Flower Extracts due to Regulation of Gut Microbiota and Short-Chain Fatty Acids in Mice Exposed to Chronic Alcohol"

_foods, 2023, doi:10.3390/foods12071428_

Round 1
Reviewer 1 Report
reduce plagiarism
Author Response
I'm sorry and it has been modified in my own language. Please review it again, thank you.
Reviewer 2 Report
The manuscript entitled “Alleviation of alcoholic fatty liver by Dendrobium officinale flower extracts due to regulation of gut microbiota and short-chain fatty acids in mice exposed to chronic alcohol”by Zhang and co-authors have developed an alcohol fatty liver diseases (AFLD) model and evaluated the preotective effect of Dendrobium officinale flower (DOF) ethanol extract on AFLD. Experimental results suggested that DOFE enhance the alcohol metabolism and resotre hepatic antioxidant capacity, and reduce the inflammatory factors levels. Authors have explored the protective effects of DOF ethanol extract on AFLD. Overall, the manuscript is well written and all the figures are self explanatory. Here are my queries/suggestions for the manuscript:-
1. Manuscript need to screen for typos grammatical errors.
2. Authors need to discuss the outcomes of latest studies.
3. What are the other protective effects of plant extract reported for AFLD.
4. Abstract need to refine.
5. In section 2.5. Analysis of hepatic biochemical indications, authors written that “homogenized in 9 mL PBS buffer at pH 7.2−7.4.” Authors should specify the specific pH of buffer instead of range of pH.
Author Response
We would like to thank you for your careful reading, helpful comments, and constructive suggestions, which has significantly improved the presentation of our manuscript.
We have made correction according to the Reviewer’s comments:
1.I've screened for typos, grammatical errors.
2.Added a discussion of the latest research results.
3.Added the other protective effects of plant extract reported for AFLD.
4.Refined the abstract.
5.The ph was determined.
Reviewer 3 Report
The manuscript delivers valuable data on alcoholic fatty liver disease treatment possibilities using flowers of Dendrobium officinale Kimura & Migo. The in vivo experiments showed that the DO extract treatment reduced fatty changes formation in liver tissue and countered enzymatic/inflammatory factors associated with liver damage. The authors did also assess the activity in the scope of the gut microbiota composition and SCFA production, expecting beneficial outcomes, and correlating SCFA levels with taxa abundance. That is, from the reviewer’s perspective, the most important factor resulting in the novelty of the obtained results.
General comments:
The 10.1155/2020/1421853 (doi) is the first hit when doing a literature search about the manuscript keywords and presents a dataset useful for comparison/association with the author’s results. I would like to ask the authors if they considered discussing their results with the outcomes obtained in the referenced article.
The reviewer is aware that the authors have also published an extensive phytochemical screening of DO (10.1002/fsn3.2602). For the reproducibility of the results, it might be beneficial, if the authors could provide LC-MS data of the exact DO extract that was used in the in vivo experiments, using their previous work as the reference. Just a basic LCMS profile, preferably, as supplementary material.
Specific comments:
Line 2: The reviewer did find a current species name Dendrobium officinale Kimura & Migo in the WFO Plant List (http://www.worldfloraonline.org/taxon/wfo-0000940031). If the authors confirm that this is the right classification, please change it in the introduction and other key sections where the species name is introduced.
Line 20, 52, 61, 83, 257, 273: Please consider using the term microbiota/gut microbiota. Microflora/gut microflora/intestinal flora are deprecated, as there were some changes in taxonomies and there are no usual residents of Kingdom Plantae in human guts (as commensal organisms).
Line 85: Please provide the information about the source of the plant material and the identification reference.
Line 151: “…tags were filtered according to the filtering criteria in the literature.” Reference, please.
Line 154: Please correct the reviewer if he is wrong, but there seem to be no Tax4Fun results presented in the manuscript.
Line 154: Missing the reference to LEfSe LDA.
Line 156, 157: The information about the LCMS seems to be wrong. Please contact your contractor for the details about the MS settings and the analysis details.
Line 226, 247, 294: Please describe the significance indicators (*/#) in the captions of every figure, where it is needed, not only Figure 6.
Author Response
We would like to thank you for your careful reading, helpful comments, and constructive suggestions, which has significantly improved the presentation of our manuscript.
We have made correction according to the Reviewer’s comments:
1.The plant materials selected in our articles are consistent with the animal models, but the emphasis of the methods is different. The data set of the other side is not the core focus in this paper, so we did not take too much consideration. Thank you for your suggestions.
2.We will upload this section to the supplementary material later.
3.Line2: Changed it into“Dendrobium officinale Kimura & Migo”
Line 20, 52, 61, 83, 257, 273: Changed these into “gut microbiota”
Line 85: Added the information about the source of the plant material and the identification reference.
Line 151: Added the Reference.
Line 154:Added the Reference, deleted Tax4Fun.
Line 156, 157: Added the details about the MS settings and the analysis details.
Line 226, 247, 294: Added the significance indicators (*/#).
Reviewer 4 Report
Dear,
This is interesting work about Dendrobium officinale flower extracts (DOFE) supplementation in chronic alcohol-exposed mice. The experiments are well done and presented. I have minor comments:
1. What is on X-axis in the Figure 1?
2. Please provide better resolution of Figure 5.
Author Response
We would like to thank you for your careful reading, helpful comments, and constructive suggestions, which has significantly improved the presentation of our manuscript.
We have made correction according to the Reviewer’s comments:
1.X-axis in Figure 1 has been illustrated.
2.Have tried to modify.